# A Novel Approach to Assessing In-Hospital Mortality After On-Pump Aortic Valve Replacement

**DOI:** 10.3390/life15111696

**Published:** 2025-10-31

**Authors:** Anca Drăgan, Adrian Ştefan Drăgan, Ovidiu Ştiru

**Affiliations:** 1Department of Cardiovascular Anaesthesiology and Intensive Care, Prof. Dr. C.C. Iliescu Emergency Institute for Cardiovascular Diseases, 258 Fundeni Road, 022328 Bucharest, Romania; 2Faculty of General Medicine, Carol Davila University of Medicine and Pharmacy, 8 Eroii Sanitari Blvd, 050474 Bucharest, Romania; adrian-stefan.dragan2023@stud.umfcd.ro (A.Ş.D.); ovidiu.stiru@umfcd.ro (O.Ş.); 3Department of Cardiovascular Surgery, Prof. Dr. C.C. Iliescu Emergency Institute for Cardiovascular Diseases, 258 Fundeni Road, 022328 Bucharest, Romania

**Keywords:** cardiac surgery, in-hospital mortality, platelet-to lymphocytes ratio, postoperative management, vasoactive-inotropic score

## Abstract

Background: Surgical aortic valve replacement (SAVR) is the main treatment for severe aortic valve disease, the most common valvular heart disease worldwide. Methods: We evaluated the in-hospital mortality risk factors and predictors following on-pump SAVR. We retrospectively reviewed data from consecutive patients treated at a tertiary center from 2022 to 2024, focusing on routine hematological data and inflammatory indexes, alongside established factors. Results: Postoperative vasoactive-inotropic score (VIS) (OR 1.058, CI 95%: 1.007–1.112), platelet count (OR 1.033, CI 95%: 1.002–1.064), lymphocyte counts (OR 3.532, CI 95%: 1.507–8.278), and perioperative fresh frozen plasma transfusion (OR 1.335, CI 95%: 1.068–1.669) were independent risk factors for SAVR in-hospital mortality. VIS best predicted the endpoint (AUC 0.929, *p* = 0.001). Postoperative platelet count and platelet-to-lymphocytes ratio (PLR) outperformed the additive EuroSCORE in predicting the outcome, but not EuroSCORE II. Conclusions: Although EuroSCORE II remained superior to inflammatory indexes in predicting in-hospital death, the dynamic postoperative monitoring provided added value beyond static preoperative risk scores. This dynamic approach supported personalized monitoring and targeted therapeutic interventions. Postoperative VIS, platelet, lymphocyte counts, and PLR represent dynamic, low-cost predictors of in-hospital mortality after on-pump SAVR, offering a complementary value to EuroSCORE II–based models.

## 1. Introduction

Valvular heart disease (VHD) is an increasing cause of cardiovascular morbidity and mortality worldwide, particularly in developed countries with aging populations. This issue requires careful attention from healthcare professionals and researchers. Notably, aortic valve diseases (AVD) are responsible for 61% of all VHD deaths [1]. Among these, aortic stenosis (AS) stands as the most prevalent form of VHD in developed countries [2]. AS must be seen as a disease both of the valve and the myocardium, characterized by fibrosis and calcification of valve leaflets, progressive left ventricular (LV) hypertrophy, and myocardial fibrosis [3]. Aortic regurgitation (AR) occurs about two to three times less frequently than AS in the general population, ranking as the third most common valvular heart disease [4]. Its prevalence and severity tend to rise with age; however, it often remains underdiagnosed until a significant LV dysfunction accompanied by a decline in ejection fraction occurs [4].

Transcatheter aortic valve implantation (TAVI) represents a promising alternative in the treatment of severe aortic stenosis; however, its application in severe aortic regurgitation is still in the preliminary stages of development. In contrast, SAVR continues to be the established primary treatment for severe AVD. Its mortality rate, ranging from 1.1% to 13.9%, is significantly influenced by factors such as the patient’s comorbidities [5,6], their symptomatic status [5], the procedural volume of the medical center [7], and whether concomitant procedures are performed [6,8,9,10,11,12].

AV degeneration is linked to inflammation, atherosclerosis, and calcification, which can lead to functional issues such as AV stenosis and regurgitation [13]. Calcific AVD is a multifaceted condition marked by extensive inflammation, which is driven by abnormal paracrine and autocrine nitric oxide signaling, the presence of inflammatory adhesion molecules, and the generation of pro-inflammatory reactive oxygen species that are harmful to tissues [14]. In stenotic valves, macrophages exhibit increased inflammatory activity, leading to a shift in the population balance toward pro-inflammatory cells [15,16]. In comparison to AS valves, nonrheumatic AR valves demonstrate a significantly reduced degree of inflammatory cell infiltration and apolipoprotein deposition [17].

The inflammatory response during cardiac surgery is mainly affected by factors like tissue trauma, hypoperfusion, ischemia-reperfusion injury, and the use of cardiopulmonary bypass (CPB) [18]. The key components of the CPB-associated inflammatory response encompass, but are not limited to, the activation of the complement system, the release of cytokines, the pro-inflammatory cascade, the generation of nitric oxide, the coagulation–fibrinolytic system, and both cellular and immune responses [19]. Furthermore, this response involves the activation of the endothelium [19]. The systemic inflammatory response syndrome is related to CPB, with reported incidence rates ranging from 20% to 58.7% [20]. This condition can progress to multi-organ dysfunction syndrome and even result in death. Therefore, it is important to evaluate the inflammatory status to better assess patient outcomes.

Biomarkers, which can be costly and frequently difficult to obtain, are utilized to assess a patient’s inflammatory status during the perioperative period. Blood count analysis and inflammatory indexes can serve as more accessible and complementary tools to EuroSCORE II in assessing perioperative risk. Together, these methods provide valuable insights that can enhance the optimization of personalized treatment approaches.

Our retrospective study aimed to provide clinicians with some cost-effective tools to enhance perioperative risk stratification and enable personalized monitoring. This approach might ensure the timely application of both preventive and therapeutic interventions, ultimately improving overall patient outcomes. Our primary objective was to assess the risk factors and predictors associated with in-hospital mortality following on-pump SAVR. Secondly, we investigated the in-hospital death predictors specifically in patients undergoing on-pump SAVR for severe AS. Thirdly, we focused on the role of blood cell counts and inflammatory indexes in this clinical context, employing a perioperative dynamic analysis of these values. Our study explored the potential of perioperative blood cell counts and inflammatory indexes as low-cost, noninvasive prognostic indicators in this setting, while also comparing them to established variables, including the additives EuroSCORE and EuroSCORE II. 

## 2. Materials and Methods

### 2.1. Study Design

In-hospital mortality, defined as the death occurring during the same hospitalization after surgery, was our predefined study endpoint. Our main objective was to identify in-hospital mortality risk factors and predictors associated with on-pump SAVR. Secondly, we focused our analysis regarding the in-hospital mortality predictors in the subgroup of patients that were previously diagnosed with severe AS who underwent on-pump SAVR. Additionally, we analyzed the dynamics and significance of the blood cell count and inflammatory indexes within this clinical context.

To fulfill the objectives of this study, we conducted a retrospective single-center analysis. This retrospective study was conducted following the Declaration of Helsinki and was approved by the Ethics and Studies Approval Committee of the Prof. C.C. Iliescu Emergency Institute for Cardiovascular Diseases, Bucharest (No. 16112/ 1 July 2025). Patient consent was waived due to the retrospective design of the study.

The sample size was not predetermined. Rather, all eligible participants who underwent on-pump SAVR within a specified two-year timeframe were included. This consecutive selection approach minimized the potential selection bias.

We reviewed the preoperative, intraoperative, and postoperative medical records of the severe AVD consecutive patients who underwent on-pump SAVR at our tertiary center, the ‘Prof. C.C. Iliescu’ Emergency Institute for Cardiovascular Diseases, Bucharest, Romania, between 2022 and 2024. The on-pump SAVR may be performed as a single procedure or as part of a more intricate cardiac surgical intervention (complex surgery) in patients whose primary condition was attributed to the severity of aortic valve disease, specifically stenosis or regurgitation. In these complex surgeries, SAVR was performed alongside various procedures: aortic root enlargement (ARE), ascending aorta surgery (AAS), coronary artery bypass graft (CABG), mitral valve (MV) repair/replacement, and tricuspid valve (TV) repair. We excluded the patients with iterative or emergency surgery, patients with acute endocarditis, Bentall surgery, and chronic hemodialysis from our analysis, as well as the patients whose data was incomplete. We applied listwise deletion as the method of handling missing data, and analyzed the bias introduced by this method.

To achieve our primary objective, we divided the study population, the SAVR group, into two subgroups: non-survivors and survivors. The subgroup of patients previously diagnosed with severe AS who underwent on-pump SAVR will be referred to the AS_SAVR subgroup in our research.

### 2.2. Procedure

The patients had previously been diagnosed with AVD and met the criteria for surgery. The decision to proceed with surgery was made by a multidisciplinary team, which included a cardiac surgeon, a cardiologist, and a cardiac anesthesiologist. Any additional medical conditions were evaluated through a multidisciplinary approach, following our institutional protocols. The preoperative evaluation encompassed a comprehensive range of assessments, including blood tests, transthoracic echocardiography, transesophageal echocardiography, coronary angiography, arterial Doppler studies, and screenings for both viral and bacterial infections, among other methodologies. All surgeries were performed using a full sternotomy in conjunction with normothermic CPB. General anesthesia was administered, complemented by both standard and invasive monitoring techniques. Continuous invasive arterial pressure was measured, using an arterial catheter. After the anesthesia induction, a central venous catheter and a urinary catheter were placed. Intraoperative transesophageal echocardiography was performed in all patients. Following the surgical procedure, patients were subsequently monitored and cared for in the intensive care unit (ICU) to ensure optimal recovery.

### 2.3. Data Collection

We collected preoperative, intraoperative, early postoperative at ICU admission, and day-one after-surgery medical data. The studied hematologic data in the three perioperative moments were the count of the leukocytes (L), neutrophils (N), lymphocytes (Lf), monocytes (M), platelets (P), red cell distribution width—standard deviation (RDW-SD), platelet distribution width (PDW), the mean platelet volume (MPV), and the inflammatory indexes: systemic inflammatory response index (SIRI), systemic inflammatory index (SII), aggregate index of systemic inflammation (AISI), neutrophils to lymphocyte ratio (NLR), monocytes to lymphocyte ratio (MLR), and platelet to lymphocyte ratio (PLR). We retrospectively calculated these indexes [21]:SIRI = N × M/Lf; SII = N × P/Lf; AISI = N × P × M/Lf; NLR = N/Lf; MLR = M/Lf; PLR = P/Lf.

We used the additive EuroSCORE and EuroSCORE II as the risk scores to assess the mortality risk.

The preoperative studied variables included biological sex, age, preoperative creatinine clearance (Clear_preop_creat), body mass index (BMI), other comorbidities (diabetes mellitus, arterial hypertension, chronic obstructive pulmonary diseases), AVD type (stenosis or regurgitation), presence of bicuspid valve, preoperative left ventricle ejection fraction (LVEF), preoperative atrial fibrillation (Preop AF), and various hematological variables (preoperative hemoglobin concentration—Hb_preop, and the data described above).

During the intraoperative phase, we reviewed the type of surgery performed (either single SAVR or a more complex procedure), the type of prosthesis used (bioprosthesis or mechanical), and the length of the surgery (Intraop_time), as well as the duration of CPB (CPB_time) and aortic cross-clamping (ACC_time).

Postoperatively, we collected data related to hematological patterns (early postoperative and on day one after surgery), early postoperative vasoactive-inotropic score (VIS), the hemostasis reintervention, and acute kidney injury (AKI) occurrence. We used the following VIS formula: VIS = dopamine dose (mcg/kg/min) + dobutamine dose (mcg/kg/min) + 100 × epinephrine dose (mcg/kg/min) + 10 × milrinone dose (mcg/kg/min) + 10,000 × vasopressin dose (U/kg/min) + 100 × norepinephrine dose (mcg/kg/min) [22]. Data recorded early postoperatively refers to information collected upon admission to the ICU. We also reviewed the data concerning perioperative fresh frozen plasma (FFP) and red blood cell (RBC) transfusion.

### 2.4. Statistical Analysis

We used SPSS 30, applying a 95% threshold for statistical significance (*p* ≤ 0.05). Since our data did not follow a normal distribution (Shapiro–Wilk test), we presented the quantitative variables using the median and interquartile range (IQR). The Mann–Whitney test was employed to evaluate these variables across the two subgroups. Categorical variables were presented as counts and percentages (n%) and were analyzed using Fisher’s exact test.

The inflammatory indexes (SIRI, SII, AISI, NLR, MLR, PLR) and the count of the blood cells (leukocytes, neutrophils, monocytes, lymphocytes, platelets) were graphically displayed using box plots to illustrate their trends at three time points: preoperatively, early postoperatively, and the day after surgery. Friedman’s test was used to evaluate the significance of variables’ changes through these three perioperative moments. When significant changes were detected, a post hoc analysis was conducted (Wilcoxon signed-rank test with Bonferroni correction) with a corrected *p*-value of 0.016.

Each variable was tested individually using univariable binary logistic regression. If a significant result was obtained, we assessed it for multicollinearity. Variables with variance inflation factors (VIF) that were lower than five were included in a statistical model. Furthermore, this statistical model was tested using multivariable binary logistic regression. We presented the statistical significance, the percentage of accurately classified cases, and the results of the Hosmer and Lemeshow test. This analysis aimed to identify the independent risk factors for in-hospital mortality in on-pump SAVR. We reported the odds ratio (OR), 95% confidence interval (CI 95%), and the corresponding *p*-values.

To evaluate the predictive capacity of the quantitative variables regarding the endpoint, we performed the receiver operator characteristic (ROC) analysis on the entire SAVR study population and, specifically, on the AS_SAVR subgroup. This analysis facilitated a comparison of the variables, including hematological data and inflammatory indexes, with each other and with the risk scores (additive EuroSCORE and EuroSCORE II) within the study’s context. We reported AUC, *p*, CI 95%, and the cut-off values derived from the Youden index, along with their respective sensitivity (Ss) and specificity (Sp), when AUC reached statistical significance.

## 3. Results

### 3.1. Data Presentation

A total of 616 AVD consecutive patients underwent on-pump SAVR surgery between 2022 and 2024, at our tertiary center, the Prof. C.C. Iliescu Emergency Institute for Cardiovascular Diseases, Bucharest, Romania. We excluded 194 patients from our analysis, resulting in a final cohort of 422 patients (SAVR group). The non-survivors subgroup consisted of 17 (4%) patients. No patients died during the operation or on the day of the surgery. Severe AS was identified in 352 patients (AS_SAVR group), of whom 15 (4.26%) did not survive. Figure 1 provides the diagrams of the study.

The SAVR patients were 66 [58–70] years old, with a BMI of 28.07 [24.33–32.11], and mostly men (263/422, 62.33%). Among the participants, 214 patients (50.71%) received isolated SAVR, while the remaining 208 patients (49.28%) underwent more complex on-pump cardiac surgeries. For a detailed breakdown of the surgical procedures performed on the patients included in our study, please refer to Appendix A.

Table 1 outlines the patients’ variables associated with the predefined endpoint of in-hospital death, which demonstrated statistically significant results in the Mann–Whitney/exact Fisher analysis. Furthermore, Appendix A presents the findings of this analysis for all tested variables.

#### 3.1.1. Preoperative Data

Non-survivors presented significantly higher risk scores and additive EuroSCORE and EuroSCORE II, compared to survivors (*p* = 0.001). The in-hospital mortality was significantly higher in patients who underwent complex cardiac surgery (*p* = 0.001).

There were no significant differences between survivors and non-survivors in terms of age, sex, BMI, type of AVD, Preop AF, Hb_preop, Clear_preop_creat, bicuspid AV, type of aortic prosthetic valve, or LVEF (Appendix A).

The preoperative hematologic data and the inflammatory indexes showed no significantly different values between the two subgroups (*p* > 0.05) (Appendix A).

#### 3.1.2. Intraoperative Data

Non-survivors experienced significantly longer intraoperative times, along with increased CPB_time and ACC_time (Table 1).

#### 3.1.3. Early Postoperative Data

We found higher early postoperative VIS (*p* = 0.001) in non-survivors compared to survivors. On the other hand, the postoperative hematologic data that were tested early presented significantly different values for survivors compared to non-survivors, except for postoperative PDW, MPV, and M count (Table 1). Among the tested inflammatory indexes measured at ICU admission, only PLR was significantly lower in the non-survivors compared to the survivors. Additionally, the postoperative–preoperative change in PLR value reflected the same pattern (Table 1). Regarding early postoperative PLR, we observed that non-survivors exhibited both a significantly lower postoperative platelet count and a significantly higher lymphocyte count in comparison to survivors (Table 1).

#### 3.1.4. Day One After Surgery Data

From the studied day one after surgery data, only the RDW-SD, MPV, and P count showed significantly different values in the two subgroups of patients (Table 1 and Appendix A).

#### 3.1.5. Postoperative Complications

The patients who died were more likely to develop postoperative AKI (*p* = 0.001). Survivors received significantly fewer RBC and FFP units compared to those who did not survive (Table 1). The number of reinterventions for surgical hemostasis was significantly higher in non-survivors (*p* = 0.001). Our analysis did not reveal a significant incidence of reinterventions for surgical hemostasis among patients with severe AS compared to those with severe regurgitation (*p* = 0.127, exact Fisher test).

### 3.2. The Perioperative Dynamics of the Hematological Data and Inflammatory Indexes

The patients’ hematological data and the inflammatory indexes were analyzed during three perioperative periods: preoperatively, postoperatively, and on the day after surgery. The inflammatory indexes presented significantly increasing values from preoperative to postoperative and day one after surgery in the survivors subgroup (*p* = 0.001 Friedman’s test), except for the PLR values. The statistical analysis, using Friedman’s test followed by the Wilcoxon Signed Test with Bonferroni correction (corrected *p* = 0.016), is detailed in Appendix A. Although PLR exhibited a similar trend, there was no significant difference between the postoperative and preoperative values in the SAVR patients who survived (Appendix A, Figure 2).

In non-survivors, the day-one after surgery MLR value was increased than the postoperative MLR, which was higher compared to preoperative MLR value. NLR, AISI, SIRI, and SII presented the same trend in the same subgroup, but without significance, when comparing the postoperative value to the day one after surgery value (Appendix A).

The postoperative PLR values were significantly lower compared to their preoperative level, with further increased values in day one after surgery (Figure 2). We cannot find a significant difference between the day one after surgery PLR value and the preoperative one in non-survivors (Appendix A).

We also studied the dynamics of the blood cell count (L, N, M, Lf, P) in the three perioperative moments (Appendix A).

Figure 3 presents a box plot showing the dynamics of blood cell counts during preoperative, postoperative, and one day after surgery in the survivors and non-survivors subgroups.

We observed a significant increase in the levels of L and N after surgery, followed by a notable decrease on the first day post-surgery in both survivors and non-survivors (Appendix A, Figure 3). Monocyte levels also showed a significant rise from preoperative to postoperative measurements in both subgroups. Additionally, there was a further increase in monocyte levels on the first day after surgery, although this was only statistically significant in the survivors subgroup. The lymphocyte levels significantly decreased from preoperative to postoperative measurements, as well as those on the day following surgery. This trend was only significant in the survivors group. In contrast, the non-survivors showed only a slight decrease in postoperative lymphocyte levels compared to their preoperative levels. In contrast, the postoperative platelet counts were significantly lower in both subgroups compared to their preoperative counts. However, the survivors subgroup demonstrated a significant increase in platelet count on the first day following surgery (Appendix A).

### 3.3. The Binary Logistic Regression Analysis

The univariable binary logistic regression analysis identified several variables that exhibited significant results when targeting in-hospital death (Table 2 and Appendix A).

Referring to the preoperative period, LVEF, EuroSCORE additive, and EuroSCORE II were recognized as risk factors in univariable analysis (Table 2). Complex surgery, intraoperative time, CPB_time, and ACC_time were the intraoperative factors that exhibited significant results in univariable binary logistic regression analysis (Table 2).

Among the early postoperative period variables, the leukocytes count, the neutrophils count, the postoperative platelet count, the lymphocytes count, RDW-SD, early postoperative VIS, and the requirement for hemostasis reintervention were risk factors for in-hospital death in univariable analysis. Additionally, the P count, PDW, and MPV corresponding to day one following surgery also exhibited significant results in the univariable analysis. Moreover, perioperative RBCs and FFP transfusions were risk factors in the univariable binary logistic regression analysis of the in-hospital mortality in the SAVR study population.

The values of the inflammatory indexes, along with their delta values recorded during the three perioperative moments, did not yield statistically significant results when analyzed through univariable binary logistic regression concerning in-hospital mortality in the on-pump SAVR patients (Table 2).

After testing for multicollinearity, we included the variables with a VIF of less than five in the model for multivariable analysis. Our tested model presented statistical significance (χ(11) = 82.235, *p* = 0.001; Hosmer and Lemeshow test (χ(8) = 7.304, *p* = 0.504)) and classified 97.9% of cases. The results showed that early postoperative VIS, perioperative FFP transfusion, postoperative P count, and postoperative Lf count were significant predictors of in-hospital death in the multivariable logistic regression analysis (Table 2). These variables exhibited a direct relationship with our endpoint, except for the postoperative *p* value (Table 2). Other variables included in the model did not show significant results (Table 2).

Early postoperative VIS (OR 1.058, CI 95%: 1.007–1.112, *p* = 0.024), perioperative FFP transfusion (OR 1.335, CI 95%: 1.068–1.669, *p* = 0.011), postoperative platelet (OR 1.033, CI 95%: 1.002–1.064, *p* = 0.034) and postoperative lymphocyte (OR 3.532, CI 95%: 1.507–8.278, *p* = 0.004) counts were independent risk factors of in-hospital mortality in multivariable binary logistic regression analysis in the SAVR study population (Table 2).

### 3.4. ROC Analysis for Predicting In-Hospital Death

We conducted a ROC analysis on continuous variables to assess their ability to predict in-hospital death in on-pump SAVR and, specifically in patients with severe AS, the AS_SAVR group.

#### 3.4.1. SAVR Group (422 Patients)

Table 3 only presents the results of this analysis for variables with statistically significant results. The entire analysis is detailed in Appendix A.

Among the preoperative variables evaluated, only the EuroSCORE II, additive EuroSCORE, and preoperative neutrophil count were found to be predictive of our primary endpoint: in-hospital mortality. While the preoperative risk scores achieved AUCs greater than 0.8, the preoperative neutrophil count exhibited a reduced predictive ability, with an AUC of 0.6.

In terms of intraoperative variables, factors such as the CPB duration, aortic cross-clamp time, and overall intraoperative time demonstrated strong predictive capabilities for outcomes, with AUCs exceeding 0.8. These results surpassed those of the EuroSCORE II but fell short of the early postoperative VIS.

The early postoperative variables were effective in predicting the endpoint. Among these, early postoperative platelet count and PLR demonstrated good predictive values, with an AUC of 0.739. In contrast, the RDW-SD, L, and N counts only predicted the endpoint with moderate effectiveness, yielding AUC values between 0.6 and 0.7.

On the other hand, the early postoperative VIS provided the strongest prediction of the endpoint, with an AUC of 0.929 (CI 95% 0.889–0.968, *p* = 0.001) and a cutoff value of 13.5. Its performance outperformed that of preoperative risk scores, intraoperative variables, and any inflammatory index.

From the first day after surgery, platelet count, followed by RDW-SD and MPV, showed fair predictive values for the outcome, with AUCs ranging from 0.6 to 0.7. Additionally, perioperative RBC and FFP transfusions provided strong predictions of the outcome, surpassing the predictive ability of EuroSCORE II, but remaining inferior to the early postoperative VIS.

In conclusion, SII, SIRI, AISI, NLR, and MLR did not demonstrate any significant AUC in ROC analysis in SAVR patients, regardless of the perioperative period. It is important to highlight that, in the SAVR group, the early postoperative platelets and early postoperative PLR better predicted the endpoint compared to the additive EuroSCORE, but was less accurate than EuroSCORE II (Figure 4). Furthermore, early postoperative VIS emerged as the most effective predictor of in-hospital mortality.

#### 3.4.2. AS_SAVR Group (352 Patients)

Table 4 only presents the results of this analysis for variables with statistically significant results. The entire analysis is detailed in Appendix A.

EuroSCORE II, the additive EuroSCORE, and the preoperative leukocyte and neutrophil counts were identified as predictors of the primary endpoint: in-hospital mortality. EuroSCORE II achieved AUCs exceeding 0.8; however, the preoperative neutrophil and leukocyte count demonstrated a diminished predictive capability, reflected by an AUC of 0.65–0.67. Perioperative RBC and FFP transfusions provided strong predictions of the outcome, outperforming the predictive effectiveness of EuroSCORE II but remaining less effective than the early postoperative VIS.

In examining intraoperative variables, factors like the duration of CPB and the total intraoperative time showed strong predictive abilities for outcomes, with AUC values exceeding 0.8. These results outperformed those derived from the EuroSCORE II; however, they did not achieve the predictive accuracy of the early postoperative VIS. Additionally, the aortic cross-clamp time had a predictive capacity that was lower than that of the EuroSCORE II.

Early postoperative variables measured at the time of ICU admission proved to be effective in predicting in-hospital mortality for patients undergoing on-pump AS_SAVR. Among these factors, VIS emerged as the most significant predictor, showcasing an excellent AUC of 0.943 (95% CI: 0.905–0.980). The early postoperative PLR followed closely, demonstrating a strong predictive capability with an AUC of 0.814 (95% CI: 0.701–0.927) (Figure 5).

Notably, both the postoperative PLR and the change in PLR from preoperative to postoperative measurements were more effective in predicting outcomes than the original EuroSCORE, although they did not surpass EuroSCORE II (Figure 5). Therefore, the postoperative PLR may serve as a valuable supplementary tool for outcome prediction, in conjunction with EuroSCORE II, in this context. Furthermore, early postoperative counts of platelets, leukocytes, lymphocytes, and neutrophils demonstrated strong predictive capabilities, with the area under the curve (AUC) values surpassing 0.7. In contrast, early postoperative systemic inflammation index (SII) also predicted outcomes, but had a comparatively lower AUC value.

In terms of the variables measured one day after surgery, both the platelet count and the day-one postoperative PLR delta value, along with RDW, were predictive of in-hospital mortality; however, their predictive capability was reduced, as indicated by an AUC ranging from 0.64 to 0.67.

We reaffirm that postoperative PLR and the postoperative and day-one platelet counts were inversely related to our endpoint.

## 4. Discussion

We demonstrated that VIS at ICU admission, perioperative FFP transfusion, postoperative platelets, and lymphocytes values were independent risk factors of in-hospital death in patients who underwent on-pump SAVR. VIS emerged as the most significant predictor, achieving an excellent AUC of more than 0.9. The early postoperative PLR demonstrated a strong predictive capability with an AUC of 0.814 (95% CI: 0.701–0.927) in patients with severe aortic stenosis. The result was also noted in the overall SAVR group, albeit with a lower AUC of 0.739. The postoperative platelet count showed the same result in the SAVR group. Although EuroSCORE II and other established variables, such as intraoperative and CPB time, and transfusion requirements, remained superior to inflammatory indexes in predicting the outcome, the dynamic postoperative approach could offer additional value beyond static preoperative risk scores.

We analyzed data from 422 on-pump SAVR patients. A total of 208 (49.28%) patients underwent the procedure as part of a more complex surgery. In the overall SAVR group, we found that 4% of patients died during their hospitalization, following the surgery. The in-hospital mortality rate was close to that reported in the literature.

The SAVR in-hospital mortality rate is significantly influenced by factors such as the patient’s comorbidities, their symptomatic status, the procedural volume of the medical center, and whether concomitant procedures are performed. The reported overall operative mortality rate (OOMR) ranges from 1.1% for isolated SAVR in patients with chronic severe AR [5], to 2.7% in patients with both a LVEF of less than 30% and severe LV dilation [5], to 3% when additional mitral or tricuspid valve interventions were performed [6], to 4.6% in Nam et al.’s study [7], and 13.9% in triple valve surgery [12]. Arshad et al. reported an in-hospital OOMR of 8.34% when analyzing the SAVR combined with MV surgery [11]. They noted a higher mortality rate of 9.91% for SAVR-MV replacement compared to 5.57% for SAVR-MV repair [11]. Performing ARE increased the operative mortality rate from 2.3% to 3.8% [10]. When combined with CABG, the reported OOMR was 5.1% [9]–5.9% [8]. In contrast, patients undergoing a double valve replacement with CABG had a mortality rate of 10% [8].

Many studies analyzed VIS at the end of surgery in relation to in-hospital mortality in cardiac surgery with CPB [23,24,25] and in various other surgical settings [26]. Although the ideal moment when we must calculate it is still under debate, VIS represents an easy and inexpensive score to calculate [25], with elevated VIS values associated with poor outcomes [23,24,25]. We also found early postoperative VIS calculated at ICU admission, which is an independent risk factor of in-hospital mortality in on-pump SAVR, and the best predictor of our endpoint, outperforming the EuroSCORE II.

The VIS is used to quantify the level of cardiovascular support at a specific moment, and it can be considered to be an indicator of more severe hemodynamic disorders. While these medications are necessary to prevent hypotension and hypoperfusion, which can result in organ dysfunction and death, they can also increase myocardial oxygen consumption and trigger cardiac arrhythmias. High doses of vasoactive and inotropic medications, especially catecholamines, negatively impact organ function and can lead to immune-mediated injury [27]. Moreover, they may lead to peripheral, intestinal, and cardiac ischemia, which can be fatal. Furthermore, catecholamine use was associated with immunosuppression, bacterial growth, increased bacterial virulence, biofilm formation, insulin resistance, and hyperglycemia [28]. That is why a judicious combination of vasopressors, inotropes, and volume repletion is required, based on advanced hemodynamic monitoring.

Perioperative FFP transfusion was also an independent risk factor for in-hospital death in our analysis, succeeding a good prediction of the poor outcome. Xu et al. also reported that a higher volume of perioperative FFP transfusion was associated with increased in-hospital mortality in surgical patients without massive transfusion [29]. They also found that FFP volume was significantly associated with inferior postoperative outcomes, including superficial surgical site infections, nosocomial infection, length of stay, ventilation time, and acute respiratory distress syndrome (ARDS) [29]. Similarly, elevated FFP transfusion volumes were linked to higher in-hospital mortality rates in cardiac surgery patients as well [30,31]. In the Hinton et al. multicenter study, the perioperative FFP transfusion was associated with increased 30-day mortality [32]. Furthermore, Bjursten et al. found an association between FFP transfusion and long-term mortality in patients undergoing SAVR alone or in combination with CABG [33]. Despite all this, several studies have demonstrated a lack of significant association between FFP transfusion and in-hospital mortality [34]. Instead, their findings indicated a correlation with an increased length of hospital stay [35].

Clinicians must carefully evaluate the indications for plasma transfusion. A multidisciplinary patient blood management approach is essential [36]. Guidelines support the use of perioperative point-of-care testing, which can potentially reduce the need for blood transfusions during cardiac surgery and improve clinical outcomes [36]. Additionally, prothrombin complex concentrates (PCC) can be considered as a safe and effective alternative to FFP. Compared to FFP, PCC has been associated with a significant decrease in chest tube drainage output within 24 h and a reduction in the need for RBC transfusions in patients undergoing cardiac surgery complicated by bleeding [37].

We found postoperative thrombocytopenia as an independent risk factor of in-hospital death in the SAVR group. Additionally, the postoperative P count was a better predictor of the endpoint in the SAVR group compared to the additive EuroSCORE. However, its accuracy was still lower when compared to EuroSCORE II. It is important to note that the postoperative P count presented an inverse correlation with the designated endpoint.

Studies reported that postoperative thrombocytopenia was independently associated with postoperative mortality in CPB cardiac surgery [38,39], including valvular surgery [40]. This might appear due to hemodilution, platelet activation during CPB, or platelet destruction [41]. However, Törnudd et al. showed that with moderate CPB times, platelet function might not be impaired, and no consumption of circulating platelets was detected, while only protamine administration transiently affected platelet function [42]. On the other hand, Yan et al. did not find any significant association between postoperative thrombocytopenia and in-hospital mortality in valvular surgery [43]. However, they succeeded in identifying the post-CPB thrombocytopenia risk factors—age greater than 60 years, preoperative thrombocytopenia, and CPB time—as positive predictors for postoperative thrombocytopenia, with mitral valve surgery as a negative predictor [43]. In a study by Leguyader et al., platelet activation in SAVR cases was noted with both bioprostheses and bileaflet mechanical valves, but it was absent with tilting disk mechanical valves [44].

Studies have reported that postoperative thrombocytopenia can be a complication following cardiac surgery. To help preserve platelets, strategies such as minimizing CPB time, reducing hemodilution, and modifying the CPB circuit may be beneficial. While platelet transfusions are not considered to be a viable solution for this issue, the medical literature has started to discuss this topic. Hinton et al. reported that, in perioperative bleeding in cardiac surgery patients, platelets are associated with a relative mortality benefit over FFP [31]. Yanagawa et al.’s meta-analysis found that platelet transfusion was not linked with perioperative complications in cardiac surgery patients [45]. Furthermore, Fletcher et al.’s multicentric research reported that in cardiac surgery patients, perioperative P transfusion was associated with reduced operative and 90-day mortality [35]. The authors suggested that platelet transfusion should not be intentionally avoided when weighing the risks of mortality until randomized controlled trials can either confirm or refute these findings [35].

Surgical stress and the use of CPB can lead to lymphopenia due to apoptosis [46]. Li WJ et al. demonstrated that post CPB T-cells lymphopenia might be due to higher levels of inflammatory cytokines [47]. Rodríguez-López et al. reported significant leukocytosis, neutrophilia, and lymphopenia after cardiac surgery under CPB, with a decreased CD3ζ chain expression that could be due to the increased Arg 1 activity that is secondary to the activation of neutrophils [48]. However, Castro et al. showed that the immediate postoperative lymphocytes, higher than 2175.0/mm^3^, were an indicator of poor prognosis in CPB cardiac surgery [49]. Bayer et al. found that the absolute lymphocyte count significantly decreased 48 h after surgery, with the lowest total lymphocyte levels occurring during the early postoperative period [50]. Chiarelli et al. noted that the recovery of lymphocytes in the first few days following surgery may play a crucial role in tissue repair [51]. In our study, we demonstrated that postoperative Lf level was an independent risk factor of in-hospital mortality analysis in the SAVR group.

Akdag et al. reported that PLR was higher in the severe AS group than the mild-to-moderate group, with increased PLR correlating with the severity of calcific AS [52]. Similarly, Yayla et al. found that PLR significantly increased in parallel to the severity of AS [53]. Durmaz et al. recently found no correlation between the CPB and aortic clamp times and the postoperative PLR levels in cardiac surgery, but the analysis included only CABG patients [54].

In our study, the postoperative PLR predicted in-hospital death in the on-pump SAVR study population. Additionally, the postoperative PLR (AUC 0.814, *p* = 0.001) and the preoperative–postoperative difference value of PLR (AUC 0.775, *p* = 0.001) predicted in-hospital death in the AS_SAVR group better than the additive EuroSCORE (AUC 0.753, *p* = 0.001), but less accurately than EuroSCORE II (AUC 0.847, *p* = 0.001), with postoperative PLR being inversely correlated with our endpoint.

Our findings indicated that the PLR decreased immediately after surgery, though this reduction achieved statistical significance only within the non-survivors subgroup. PLR levels dropped significantly in non-survivors upon ICU admission compared to preoperative levels, primarily due to a marked decrease in platelet count following cardiac surgery. We noted that although the preoperative platelet count was not different between the two subgroups, survivors and non-survivors, the postoperative and day one platelet count were significantly lower in non-survivors compared to survivors. Conversely, the lymphocyte count displayed only a minimal decrease in the postoperative period, relative to preoperative values. Meanwhile, other inflammatory indexes exhibited a substantial increase from preoperative to postoperative levels and increased further on the first day after surgery.

Zhou et al. recently reported the same paradoxical PLR decrease in CPB cardiovascular surgery compared to cardiovascular surgery without CPB [55]. This decrease may be attributed to the destruction of platelets associated with CPB [55]. In cardiac surgery, PLR was more related to AKI [56,57] or postoperative AF [58] than to mortality rates. In Rödel. et al.’s study, apart from PLR, all indexes were independent predictors of in-hospital mortality [59]. In the study by Shvartz et al. regarding SAVR in AS patients, SIRI was the most powerful marker of systemic inflammation, while PLR failed to obtain significant results in the univariate analysis of mortality [21].

The inflammatory indexes were also studied in other surgical settings regarding in-hospital death, such as off-pump cardiac [60], vascular [61], and thoracic surgery [62]. On the other hand, Xu et al. found that elevated PLR independently predicted both all-cause and cardiovascular mortality risks among hypertensive patients [63]. Additionally, in the analysis of Zhai et al., PLR was independently associated with an increased risk of in-hospital mortality in the cardiac ICU [64].

Actions should be taken to maintain postoperative platelet count and reduce the inflammatory response related to cardiac surgery. An expert consensus recently issued a new comprehensive approach to managing perioperative inflammation in cardiac surgery with CPB. The recommendations referred to precise temperature management, perfusion, ventilation strategies, and controlled perioperative blood transfusions [20]. Some novel pharmacological agents, such as alpha-2 agonists and serine protease inhibitors, are recommended, as well as miniaturized CPB systems with enhanced biocompatibility [20]. The consensus established distinct glycemic control ranges for diabetic and non-diabetic patients and demonstrated a distinction between infectious and non-infectious inflammatory causes [20].

Limits of the study: Our study has several limitations that should be acknowledged. First, it utilized a retrospective design, conducted at a single center, involving consecutive patients. This approach aimed to minimize potential selection bias, since the sample size was not predetermined. Secondly, the heterogeneity of our cohort may introduce further biases. The on-pump SAVR was performed either as a standalone procedure or as part of a more complex surgery in patients that were primarily affected by severe aortic valve disease, specifically stenosis or regurgitation. Additionally, the presence of missing data could lead to further biases. We addressed missing data through listwise deletion, concluding that its impact on the overall study was negligible, as only two patients were excluded due to missing information. The two subgroups, non-survivors and survivors, are unequal in size. To address this discrepancy, we primarily utilized nonparametric statistical methods, such as the Mann–Whitney test and the exact Fisher test, since most of the data did not adhere to a normal distribution. We also faced several institutional limitations that affected our analysis. These included inconsistencies in the data regarding various inflammatory indicators, such as protein C levels and fibrinogen, the types of cardioplegic solutions used, and a lack of detailed information on transfusion requirements (types of transfusions, timing, and the use of point-of-care methods). These limitations restricted our ability to perform a more comprehensive analysis. Furthermore, we lacked external validation for our findings, which should be considered when interpreting our results. To validate our findings, future prospective multicenter studies are necessary.

Clinical implications. Despite these challenges, we investigated the risk factors and predictors of in-hospital mortality in patients undergoing on-pump SAVR. Our analysis primarily focused on routine hematological data and inflammatory indexes, alongside established factors such as cardiopulmonary bypass time, aortic cross-clamping time, VIS, and transfusion requirements. Our goal was to provide clinicians with tools to assess mortality risk throughout the perioperative process as a dynamic journey, enabling personalized monitoring and targeted therapeutic actions. EuroSCORE II still remains the most important preoperative predictor of the in-hospital death. Upon ICU admission, the VIS calculation, the postoperative platelet count, and postoperative PLR may serve as valuable and easy-to-use tools for predicting in-hospital mortality in patients undergoing on-pump SAVR. VIS was the strongest postoperative predictor in our analysis. Early postoperative PLR may serve as a valuable tool in dynamic risk assessment, complementary to risk scores. Furthermore, we highlighted the significance of avoiding fresh frozen plasma administration in this clinical setting, as our findings show it to be an independent risk factor of in-hospital death. The management of platelets during surgery and the inflammatory response are crucial, as we noted that postoperative platelet levels and lymphocyte counts are associated with an increased risk of in-hospital death in on-pump SAVR patients.

To further explore the usefulness of this type of medical data in predicting mortality, prospective studies are necessary. New data from routine blood analyses may reveal other cost-effective elements that could help in risk-stratifying patients undergoing SAVR. Additionally, future studies should investigate the relationship between inflammatory indexes and serum cardiac biomarkers, particularly in the on-pump SAVR context and in oncologic patients undergoing this procedure. This research could build on the observation that elevated serum cardiac biomarkers have been noted in various clinical situations, including cancer [65,66]. The relationship between inflammatory markers, preoperative medications, and various surgical and anesthetic techniques used intraoperatively, as well as diverse strategies for cardiopulmonary bypass, should also be investigated in future research.

## 5. Conclusions

Dynamic postoperative monitoring offers valuable advantages over static preoperative risk scores, such as EuroSCORE II, in predicting in-hospital mortality following on-pump SAVR. We must highlight the significant role of VIS monitoring in perioperative clinical decision-making. Our research indicated that the early postoperative VIS was the most reliable indicator of in-hospital death, achieving an AUC greater than 0.9. Moreover, the postoperative VIS, PLR, and platelet counts serve as dynamic, cost-effective predictors of in-hospital mortality in this context, providing valuable supplementary information to EuroSCORE II-based models. Additionally, factors such as the early postoperative VIS, perioperative fresh frozen plasma transfusion, and the platelet and lymphocyte counts measured at ICU admission emerged as independent risk factors for in-hospital death in this setting.

## Figures and Tables

**Figure 1 life-15-01696-f001:**
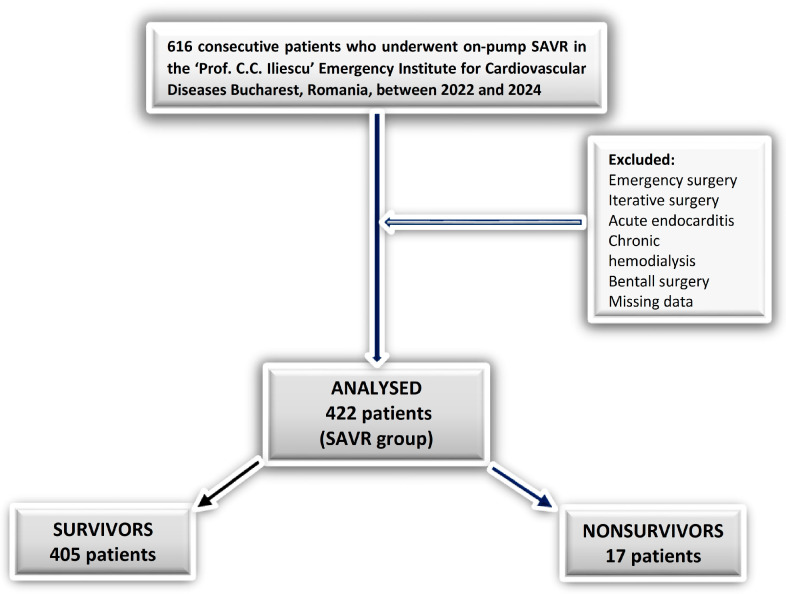
Flow diagram of patient inclusion and exclusion.

**Figure 2 life-15-01696-f002:**
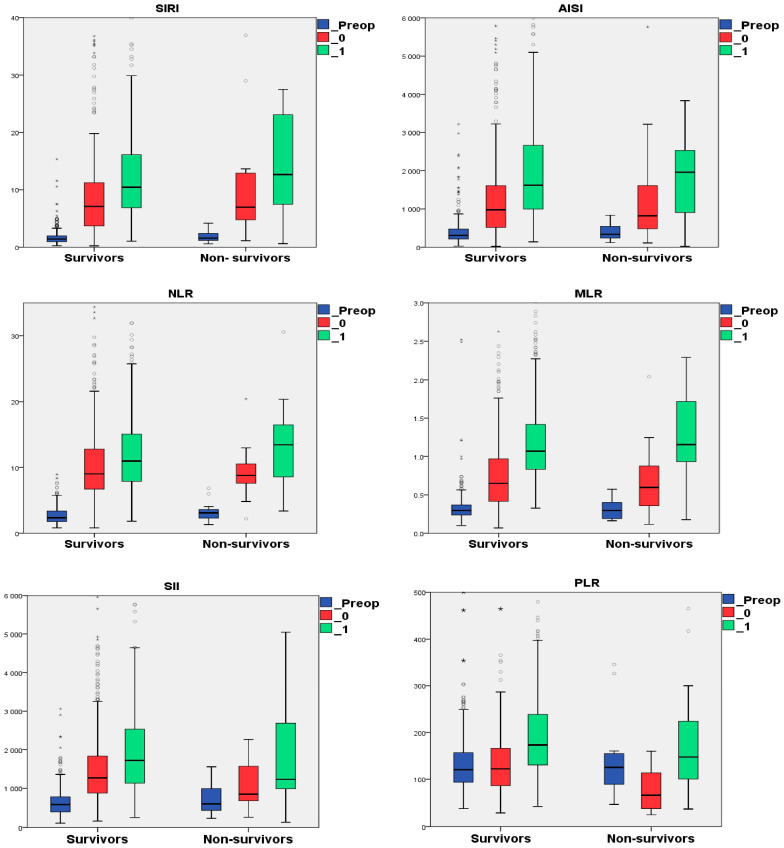
The perioperative dynamics of the inflammatory in the survivors and non-survivors subgroups. Note: _Preop refers to preoperative variable; _0 refers to early postoperative variable; and _1 refers to day one after surgery variable; “◦”represents mild outliers; “*” represents extreme outliers. Abbreviation: AISI, aggregate index of systemic inflammation; MLR, monocytes to lymphocyte ratio; NLR, neutrophils to lymphocyte ratio; PLR, platelet to lymphocyte ratio; SII, systemic inflammatory index; and SIRI, systemic inflammatory response index.

**Figure 3 life-15-01696-f003:**
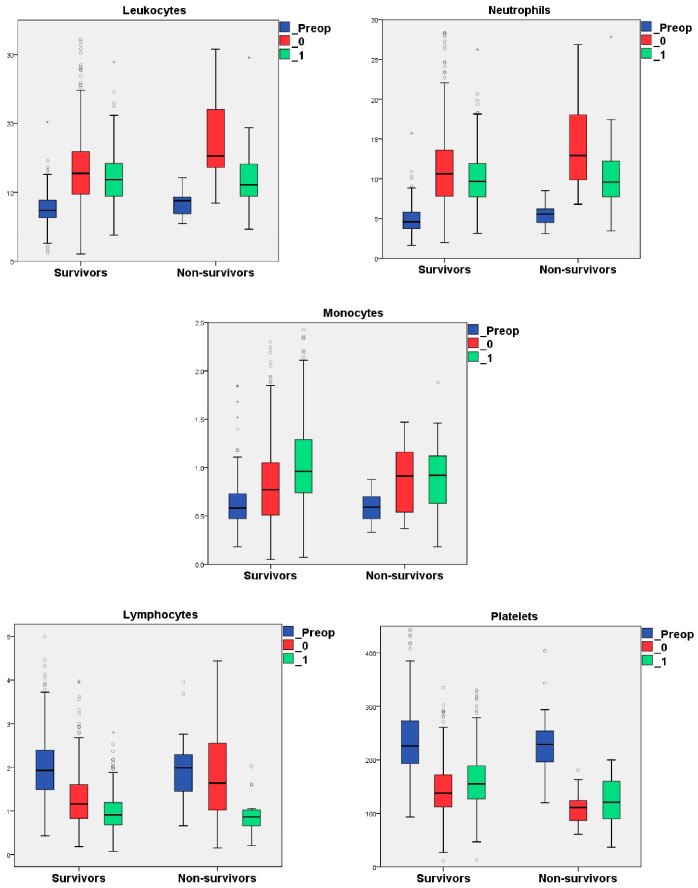
The perioperative dynamics of the blood cell counts in survivors and non-survivors subgroups. Note: “◦”represents mild outliers; “*” represents extreme outliers. Abbreviations: L, leukocytes; Lf, lymphocytes; M, monocytes; N, neutrophils; and P, platelets.

**Figure 4 life-15-01696-f004:**
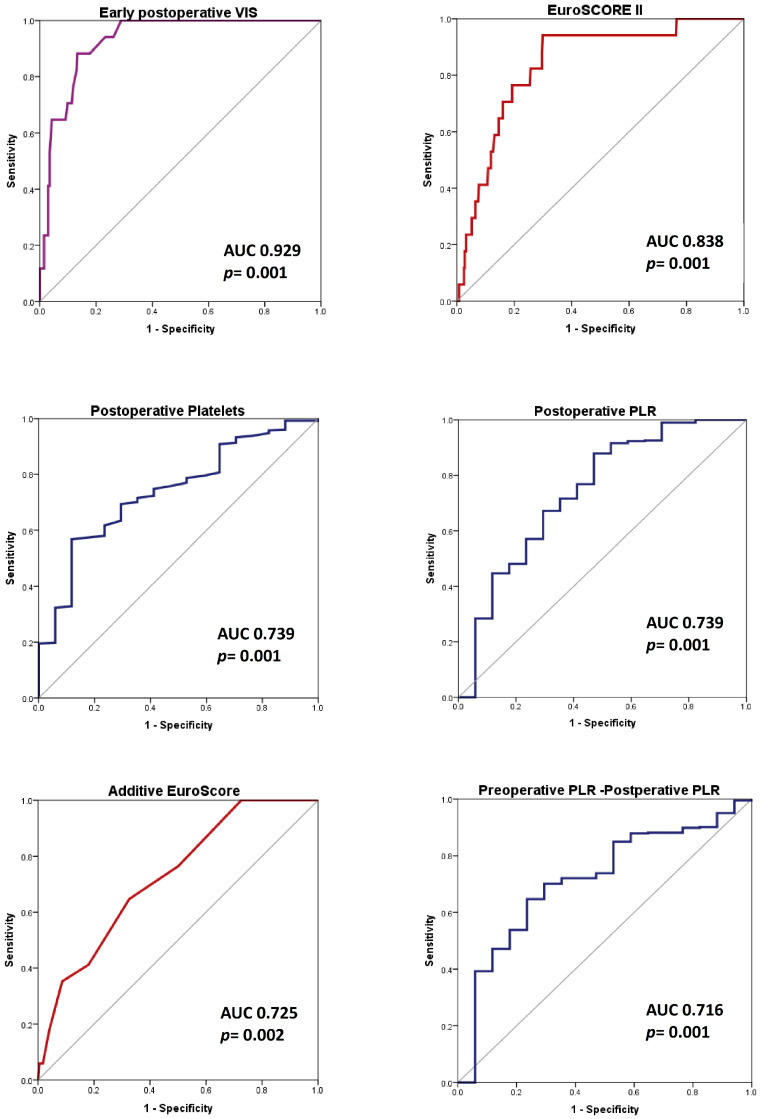
The hematological data and inflammatory indexes ROC curves, compared to risk scores and early postoperative VIS (the highest AUC in ROC analysis). Endpoint: in-hospital death endpoint in SAVR patients. Abbreviations: AUC, area under the curve; ROC, receiver operator characteristic; *p*, probability value; PLR, platelet to lymphocyte ratio; and VIS, vasoactive-inotropic score.

**Figure 5 life-15-01696-f005:**
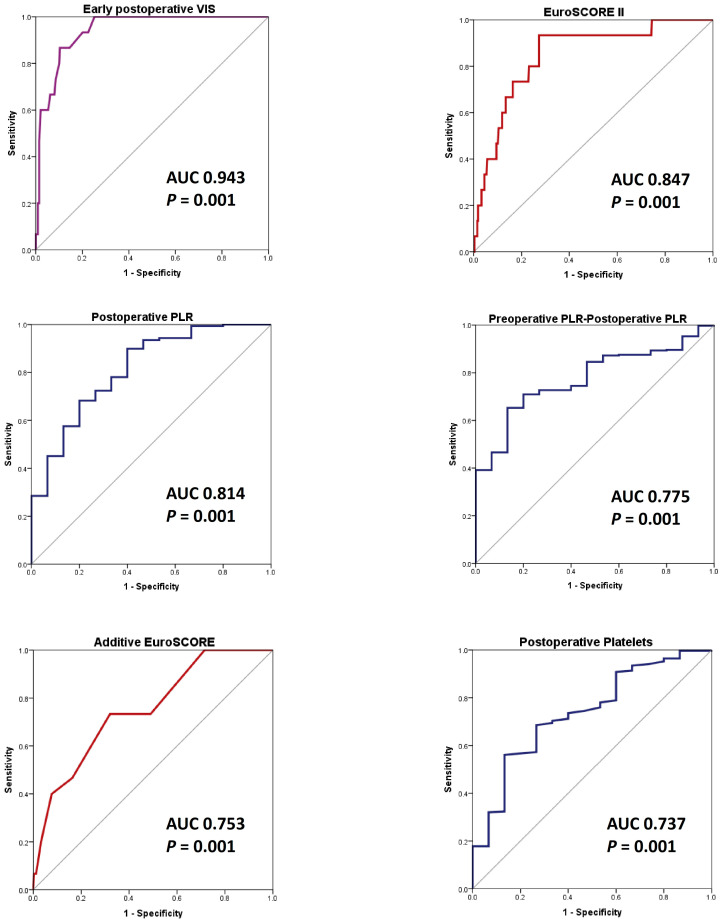
The hematological data and inflammatory indexes’ ROC curves, compared to risk scores and early postoperative VIS (the highest AUC in ROC analysis). Endpoint: the in-hospital death in AS_SAVR patients. Abbreviations: AS_SAVR, patients with severe aortic stenosis who underwent on-pump surgical aortic valve replacement; AUC, area under the curve; ROC, receiver operator characteristic; *p,* probability value; PLR, platelet to lymphocyte ratio; and VIS, vasoactive-inotropic score.

**Table 1 life-15-01696-t001:** The data reviewed in the two subgroups regarding in-hospital death endpoint (only variables with statistically significant results in Mann–Whitney/exact Fisher analysis).

Variable	Survivors (*n* = 405)	Non-Survivors (*n* = 17)	*p* ^1^
EuroSCORE ^2^	6 [4–7]	7 [5.5–9]	0.001
EuroSCORE II ^2^	1.51 [1.07–2.45]	3.4 [2.63–4.6]	0.001
Complex surgery ^3^	194 (47.9%)	14 (82.35%)	0.006
VIS ^2^	4.3 [0–9]	27 [15.5–39]	0.001
Intraop_time (h) ^2^	5 [4–5]	6 [6–8.5]	0.001
CPB_time (min) ^2^	95 [80–156.8]	174 [151.6–232]	0.001
ACC_time (min) ^2^	72 [60–90]	128 [100.5–142]	0.001
RDW-SD_0 (fl) ^2^	42.3 [40–44.75]	44.4 [41.95–48.5]	0.010
L_0 (∗10^3^/μL) ^2^	12.74 [9.74–15.92]	15.27 [12.85–22.74]	0.011
N_0 (∗10^3^/μL) ^2^	10.64 [7.8–13.59]	12.91 [9.78–18.07]	0.018
P_0 (∗10^3^/μL) ^2^	138 [112–172]	111 [86.5–127.5]	0.001
Lf_0 (∗10^3^/μL) ^2^	1.16 [0.83–1.6]	1.64 [0.98–2.58]	0.038
PLR_0 ^2^	122.5 [85.83–166.33]	65.85 [36.35–118.96]	0.001
RDW-SD_1 (fl) ^2^	43.7 [41.2–46.2]	44.5 [43.5–48.2]	0.037
MPV_1 (fl) ^2^	11.3 [10.7–12]	11.9 [11.05–12.8]	0.045
P_1 (∗10^3^/μL) ^2^	155 [127–189.5]	121 [89.5–162.5]	0.005
PLR_0-PLR_Preop ^2^	3.49 [−31.35–35.74]	−28.72 [−67.28–−8.78]	0.003
AKI ^3^	104 (25.67%)	17 (100%)	0.001
Hemostasis R ^3^	33 (8.14%)	10 (58.82%)	0.001
RBCs (units) ^2^	0 [0–2]	5 [3–6]	0.001
FFP (units) ^2^	0 [0–2]	10 [4–12]	0.001

^1^ *p* value Mann–Whitney test or exact Fisher test; ^2^ median [IQR]; ^3^ n (%). Note: “*” is a multiplication sign; _Preop, preoperative value; _0, value measured at intensive care admission; and _1, value from day one after surgery. Abbreviations: ACC_time, aortic cross clamping time; AKI, acute kidney injury; CPB, cardiopulmonary bypass; FFP, fresh frozen plasma; Hemostasis R, reintervention for hemostasis; Intraop_time, duration of the surgery; L, leukocytes count; Lf, lymphocytes count; M, monocytes count; MPV, mean platelet volume; N, neutrophils count; *p* probability value; P, platelet count; PLR, platelet to lymphocyte ratio; RBCs, red blood cell concentrates; RDW-SD, red blood cell distribution width—standard deviation; and VIS, early postoperative vasoactive-inotropic score.

**Table 2 life-15-01696-t002:** The binary logistic regression analysis targeting in-hospital death (only variables with statistically significant results).

Variable	Univariable			Multivariable	
	Exp (B)	OR (CI 95%)	*p*	OR (CI 95%)	*p*
Hemostasis R	16.1	16.1 (5.753–45.08)	0.001		0.712
Complex surgery	5.076	5.076 (1.437–17.931)	0.012		0.844
RBCs (units)	1.945	1.945 (1.530–2.473)	0.001		0.865
FFP (units)	1.486	1.486 (1.302–1.697)	0.001	1.335 (1.068–1.669)	0.011
Intraop_time (h)	2.765	2.765 (1.892–4.040)	0.001		0.115
CPB_time (min)	1.037	1.037 (1.024–1.050)	0.001		
ACC_time (min)	1.038	1.038 (1.024–1.053)	0.001		
VIS	1.069	1.069 (1.042–1.097)	0.001	1.058 (1.007–1.112)	0.024
LVEF (%)	0.939	1.064 (1.003–1.129)	0.041		
EuroSCORE	1.479	1.479 (1.78–1.856)	0.001		
EuroSCORE II	1.683	1.683 (1.334–2.124)	0.001		0.785
RDW-SD_0 (fl)	1.099	1.099 (1.014–1.193)	0.022		0.913
L_0 (∗10^3^/μL)	1.106	1.106 (1.031–1.186)	0.005		
N_0 (∗10^3^/μL)	1.104	1.104 (1.022–1.192)	0.012		
P_0 (∗10^3^/μL)	0.978	1.022 (1.008–1.036)	0.002	1.033 (1.002–1.064)	0.034
Lf_0 (∗10^3^/μL)	2.314	2.314 (1.324–4.045)	0.003	3.532 (1.507–8.278)	0.004
PDW_1 (fl)	1.176	1.176 (1.005–1.376)	0.043		
MPV_1 (fl)	1.712	1.712 (1.1–2.664)	0.017		
P_1 (∗10^3^/μL)	0.982	1.018 (1.006–1.029)	0.004		0.113

Note: “*” is a multiplication sign; _0 refers to the value of the variable measured at intensive care admission; and _1 refers to the variable value from day one after surgery. Abbreviations: ACC_time, aortic cross clamping time; CI, confidence interval; CPB, cardiopulmonary bypass; FFP, fresh frozen plasma; Hemostasis R, reintervention for hemostasis; Intraop_time, duration of the surgery; LVEF, left ventricle ejection fraction; L, leukocytes count; Lf, lymphocytes count; M, monocytes count; MPV, mean platelet volume; N, neutrophils count; OR, odds ratio; *p* probability value; P, platelet count; PDW, platelet distribution width; RBCs, red blood cell concentrates; RDW-SD, red blood cell distribution width—standard deviation; VIS, early postoperative vasoactive-inotropic score.

**Table 3 life-15-01696-t003:** ROC analysis in SAVR group (only variables with statistically significant results).

Variable	ROC			Cut off		
	AUC	*p*	CI 95%	Value	Ss (%)	Sp (%)
VIS	0.929	0.001	0.889–0.968	13.5	88.2	86.7
RBCs (units)	0.891	0.001	0.800–0.968	2.5	82.4	85.2
CPB_time (min)	0.871	0.001	0.762–0.980	160.5	76.5	91.6
FFP (units)	0.864	0.001	0.741–0.986	6.5	76.5	96.5
Intraop_time (h)	0.861	0.001	0.786–0.935	5.5	82.4	75.3
ACC_time (min)	0.839	0.001	0.740–0.938	109.5	76.5	85.7
EuroSCORE II	0.838	0.001	0.752–0.925	2.19	94.1	70.1
P_0 (∗10^3^/μL)	0.739	0.001	0.630–0.847	131.5	56.8	88.2
PLR_0	0.739	0.001	0.602–0.876	66.00	87.9	52.9
EuroSCORE	0.725	0.002	0.614–0.836	6.5	64.7	67.3
PLR_Preop-PLR_0	0.716	0.001	0.596–0.835	15.56	76.5	63.5
P_1 (∗10^3^/μL)	0.702	0.003	0.567–0.837	133.5	69.4	70.6
RDW-SD_0	0.685	0.001	0.574–0.796	40.75	100	33.6
L_0 (∗10^3^/μL)	0.682	0.010	0.543–0.820	19.55	47.1	88.9
N_0 (∗10^3^/μL)	0.669	0.016	0.532–0.807	12.23	70.6	63.7
RDW-SD_1	0.649	0.002	0.554–0.744	42.55	100	38.5
MPV_1 (fl)	0.644	0.026	0.517–0.770	10.95	88.2	37.3
N_Preop (∗10^3^/μL)	0.633	0.027	0.515–0.752	4.44	82.4	46.4

Note: “*” is a multiplication sign; _Preop refers to the preoperative value of the variable; _0 refers to the variable value measured at intensive care admission; and _1 refers to the variable value from day one after surgery. Abbreviations: ACC_time, aortic cross clamping time; AUC, area under the curve; CI, confidence interval; CPB, cardiopulmonary bypass; FFP, fresh frozen plasma; Intraop_time, duration of the surgery; L, leukocytes count; MPV, mean platelet volume; N, neutrophils count; NLR, neutrophils to lymphocyte ratio; *p* probability value; P, platelet count; PDW, platelet distribution width; PLR, platelet to lymphocyte ratio; RBCs, red blood cell concentrates; RDW-SD, red blood cell distribution width—standard deviation; Sp, specificity; Ss, sensitivity; VIS, early postoperative vasoactive-inotropic score.

**Table 4 life-15-01696-t004:** ROC analysis in AS_SAVR group (only variables with statistically significant results).

Variable	ROC			Cut off		
	AUC	*p*	CI 95%	Value	Ss (%)	Sp (%)
VIS	0.943	0.001	0.905–0.980	13.5	86.7	89.6
RBCs (units)	0.883	0.001	0.781–0.985	2.5	80	86.4
CPB_time (min)	0.870	0.001	0.748–0.991	141.5	80	88.7
Intraop_time (h)	0.867	0.001	0.789–0.945	5.5	80	78.6
FFP (units)	0.849	0.001	0.713–0.986	6.5	73.3	97
EuroSCORE II	0.847	0.001	0.752–0.941	2.19	93.3	72.8
ACC_time (min)	0.837	0.001	0.726–0.947	109.5	73.3	88.4
PLR_0	0.814	0.001	0.701–0.927	66	89.9	60
PLR_Preop-PLR_0	0.775	0.001	0.685–0.865	15.56	86.7	65.3
EuroSCORE	0.753	0.001	0.632–0.874	6.5	73.3	68
P_0 (∗10^3^/μL)	0.737	0.001	0.614–0.861	131.5	56.1	86.7
L_0 (∗10^3^/μL)	0.729	0.001	0.592–0.866	19.53	53.3	88.7
Lf_0 (∗10^3^/μL)	0.725	0.001	0.590–0.859	1.63	60	77.4
N_0 (∗10^3^/μL)	0.710	0.002	0.574–0.846	15.72	53.3	85.5
RDW-SD_0 (fl)	0.691	0.001	0.576–0.807	40.75	100	33.5
P_1 (∗10^3^/μL)	0.676	0.023	0.525–0.828	133.5	68	66.7
N_Preop (∗10^3^/μL)	0.676	0.003	0.560–0.792	4.44	93.3	44.6
SII_0	0.660	0.026	0.519–0.800	821.03	79.2	53.3
L_Preop (∗10^3^/μL)	0.658	0.017	0.529–0.788	8.13	73.3	60.2
RDW-SD_1 (fl)	0.646	0.005	0.544–0.748	42.55	100	38.9
PLR_1-PLR_0	0.645	0.014	0.530–0.759	42.31	80	49.6

Note: “*” is a multiplication sign; _Preop refers to the preoperative value of the variable; _0 refers to the variable value measured at intensive care admission; and _1 refers to the variable value from day one after surgery. Abbreviations: ACC_time, aortic cross clamping time; AUC, area under the curve; BMI, body mass index; CI, confidence interval CPB, cardiopulmonary bypass; FFP, fresh frozen plasma; Intraop_time, duration of the surgery; L, leukocytes count; Lf, lymphocytes count; N, neutrophils count; *p* probability value; P, platelet count; PLR, platelet to lymphocyte ratio; RBCs, red blood cell concentrates; RDW-SD, red blood cell distribution width—standard deviation; SII, systemic inflammatory index; Sp, specificity; Ss, sensitivity; and VIS, early postoperative vasoactive-inotropic score.

## Data Availability

The data presented in this study are available on request from the corresponding author, due to privacy and ethical restrictions.

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
