# Peer review of "A Novel Approach to Assessing In-Hospital Mortality After On-Pump Aortic Valve Replacement"

_life, 2025, doi:10.3390/life15111696_

Round 1

Reviewer 1 Report

Comments and Suggestions for Authors

The authors of this article retrospectively analyzed a 2-year experience with aortic valve surgery at a large hospital in Bucharest.
The aim of the study was to identify predictors and risk factors for mortality after aortic valve replacement.

There are several key caveats to this study:
1) There were 17 patients who died. However, only 15 of these patients had severe aortic valve disease. This raises the question of the appropriateness of performing surgery in patients with mild aortic stenosis. These patients likely had indications for other procedures, such as coronary artery bypass grafting (CABG) or mitral valve replacement, but then why include them in the analysis?
2) Overall, including patients with such a diverse range of surgical procedures in the analysis appears questionable. For example, simultaneous replacement of both the aortic and mitral valves is inherently an extremely high-risk surgical procedure and a priori leads to a more difficult postoperative period. One patient underwent SAVR+ARE+MV+TV simultaneously, and this cannot be compared with 214 patients with isolated aortic valve replacement.
3) Table 1 does not indicate the percentages of patients with COPD and DM. Surgery time is given in hours, although it would be more logical to compare it in minutes, which would also likely reveal a significant difference.
4) The compared groups are too unequal in size, which undermines the reliability of the obtained results for a large sample.
5) The results indicate that the VIS score was the most important factor in in-hospital mortality, which is obvious, since this indicator reflects the patient's severity in the early postoperative period. In my opinion, it would be much more interesting to analyze what exactly led to such an increase in the VIS score, rather than using it as a risk factor.

Therefore, the study has a number of significant shortcomings and cannot be recommended for publication in its current form.

Reviewer 2 Report

Comments and Suggestions for Authors

The manuscript addresses an important and clinically relevant topic, offering valuable insights into dynamic predictors of in-hospital mortality after on-pump SAVR. However, on my opinion, the paper requires substantial revision to improve clarity, focus, and methodology:

  1.    Define how and when the vasoactive–inotropic score (VIS) was measured, explain handling of missing data, and specify what constitutes “complex surgery.”
  2. Condense tables and figures to highlight only statistically and clinically relevant predictors; remove redundant or non-significant data.
  3. Focus on interpretation and clinical implications rather than repeating results; discuss biological mechanisms and study limitations more critically.
  4. Simplify numerical reporting, include study design and sample size, and conclude with clear clinical relevance.
  5. Acknowledge key limitations: retrospective design, single-center data, potential selection bias, lack of external validation, and missing data.
  6. Potential confounders (transfusions, surgical complexity, institutional factors) are not adequately discussed.

Reviewer 3 Report

Comments and Suggestions for Authors

This manuscript addresses a relevant and timely topic — predictors of in-hospital mortality after on-pump SAVR. The study is well designed and provides meaningful insight by comparing VIS, PLR, and hematologic markers with established EuroSCORE models. The results are clinically valuable and clearly presented.

Some revisions are recommended to improve clarity and overall flow. The Introduction could be more concise and better connected to the study hypothesis. In the Methods, please define “complex surgery” earlier, add references for calculated indices, and briefly justify the sample size. In the Results, avoid repeating numerical data, simplify figure captions, and emphasize that VIS was the strongest postoperative predictor while PLR complemented EuroSCORE II.

The Discussion would benefit from a short summary of main findings and a clearer mechanistic interpretation of key results, including why PLR decreased in non-survivors. Consider expanding the Limitations section to note the single-center, retrospective design and lack of external validation. The Conclusions could briefly restate that dynamic postoperative markers provide additional value beyond static preoperative risk scores.

Language and references are generally good but should be reviewed for consistency and recent updates. With these refinements, the manuscript would be clearer, more balanced, and scientifically stronger.

Round 2

Reviewer 1 Report

Comments and Suggestions for Authors

Dear Dr. Anca Drăgan
You and your colleagues have done fantastic work assessing risk factors for adverse outcomes in patients who have undergone SARV surgery. I see many positive improvements in the revised version of the article, making the work clearer and more logical. However, it remains unclear to me how we can include patients with various aortic valve pathologies in the overall analysis. In particular, you included both patients with aortic stenosis and aortic regurgitation in the overall analysis. Consider, for example, the issue of postoperative bleeding. Patients with advanced aortic stenosis are characterized by the development of so-called Hyde syndrome, caused by mechanical destruction of von Willebrand factor as it passes through the narrowed aortic annulus. This phenomenon is not observed in aortic insufficiency. Accordingly, we cannot include both patient groups in the overall analysis, since patients with aortic stenosis have a higher risk of bleeding. And in general, in my opinion, your analysis should only concern patients with isolated aortic stenosis.

Author Response

Comments and Suggestions for Authors

Dear Dr. Anca Drăgan

You and your colleagues have done fantastic work assessing risk factors for adverse outcomes in patients who have undergone SARV surgery. I see many positive improvements in the revised version of the article, making the work clearer and more logical. However, it remains unclear to me how we can include patients with various aortic valve pathologies in the overall analysis. In particular, you included both patients with aortic stenosis and aortic regurgitation in the overall analysis. Consider, for example, the issue of postoperative bleeding. Patients with advanced aortic stenosis are characterized by the development of so-called Hyde syndrome, caused by mechanical destruction of von Willebrand factor as it passes through the narrowed aortic annulus. This phenomenon is not observed in aortic insufficiency. Accordingly, we cannot include both patient groups in the overall analysis, since patients with aortic stenosis have a higher risk of bleeding. And in general, in my opinion, your analysis should only concern patients with isolated aortic stenosis.

Response

Dear Reviewer,

Thank you for your valuable suggestions and comments, as well as your kind words about our manuscript! We truly appreciate the time and effort you have dedicated to evaluating our work.

Our cohort is indeed heterogeneous. We included patients who underwent on-pump SAVR performed as a single procedure or as part of a more intricate cardiac surgical intervention (complex surgery) in patients whose primary condition was attributed to the severity of aortic valve disease, specifically stenosis or regurgitation.

We explicitly mentioned this limitation in the study presentation.

Our main objective was to identify in-hospital mortality risk factors and predictors associated with on-pump SAVR in general. Secondly, we focused our analysis on the in-hospital mortality predictors in the subgroup of patients previously diagnosed with severe aortic stenosis who underwent on-pump SAVR (please see Section 3.4).

Regarding the differences in bleeding complications associated with surgical aortic valve replacement in cases of aortic stenosis versus regurgitation, our analysis did not reveal a significant incidence of reinterventions for surgical hemostasis among patients with severe aortic stenosis compared to those with severe regurgitation (p = 0.127, exact Fisher test). Based on your feedback, we included this analysis in the manuscript (Section 3.1, Postoperative complications).

Reviewer 2 Report

Comments and Suggestions for Authors

The revised manuscript satisfactorily addresses all major concerns raised in my initial review. The responses are clear, detailed, and the changes substantially improve the clarity, methodology, and overall quality of the paper.

Author Response

Comments and Suggestions for Authors

The revised manuscript satisfactorily addresses all major concerns raised in my initial review. The responses are clear, detailed, and the changes substantially improve the clarity, methodology, and overall quality of the paper.

 Response

We thank you for your useful suggestions and comments and your kind words about our manuscript! We greatly appreciate the time and efforts you have taken for the evaluation of our work!

Reviewer 3 Report

Comments and Suggestions for Authors

The revised version of the manuscript shows clear and substantial improvement in structure, clarity, and clinical focus. The topic is clinically meaningful and the study is well-designed, providing valuable insights into perioperative mortality risk stratification following on-pump aortic valve replacement. By integrating both traditional (EuroSCORE II) and dynamic postoperative markers (such as VIS and inflammatory ratios), the authors successfully bridge established and emerging risk assessment approaches, which enhances the translational value of the work.

The manuscript now reads fluently, with transparent methodology and a coherent, well-referenced discussion. The addition of the Clinical Implications and Limitations sections notably strengthens the overall interpretation and scientific maturity of the paper.

Only a few minor editorial refinements are suggested. The quotation marks around the institution’s name (“Prof. Dr. C.C. Iliescu”) should be removed throughout the text and affiliations. In the Keywords section, the order of “platelet-to-lymphocytes ratio” and “postoperative management” may be switched to maintain strict alphabetical order. Finally, adding one concise sentence in the Conclusions to emphasize the potential role of VIS monitoring in perioperative clinical decision-making would further strengthen the practical message of the study.

Additionally, a minor editorial note: in Figure 1, the caption text appears to have been replaced or shifted by the figure label — this likely requires correction during production. Also, for stylistic consistency, the terms “survivors” and “non-survivors” could be written in lowercase throughout the manuscript.

Overall, this is a well-written, methodologically sound, and clinically valuable manuscript. The authors have clearly taken the reviewers’ feedback into account and refined the paper with care and precision. The study will be of clear interest to clinicians and researchers in cardiac surgery and perioperative medicine. I recommend acceptance after these very minor editorial adjustments.

Author Response

Comments and Suggestions for Authors

The revised version of the manuscript shows clear and substantial improvement in structure, clarity, and clinical focus. The topic is clinically meaningful and the study is well-designed, providing valuable insights into perioperative mortality risk stratification following on-pump aortic valve replacement. By integrating both traditional (EuroSCORE II) and dynamic postoperative markers (such as VIS and inflammatory ratios), the authors successfully bridge established and emerging risk assessment approaches, which enhances the translational value of the work.

The manuscript now reads fluently, with transparent methodology and a coherent, well-referenced discussion. The addition of the Clinical Implications and Limitations sections notably strengthens the overall interpretation and scientific maturity of the paper.

Only a few minor editorial refinements are suggested. The quotation marks around the institution’s name (“Prof. Dr. C.C. Iliescu”) should be removed throughout the text and affiliations. In the Keywords section, the order of “platelet-to-lymphocytes ratio” and “postoperative management” may be switched to maintain strict alphabetical order. Finally, adding one concise sentence in the Conclusions to emphasize the potential role of VIS monitoring in perioperative clinical decision-making would further strengthen the practical message of the study.

Additionally, a minor editorial note: in Figure 1, the caption text appears to have been replaced or shifted by the figure label — this likely requires correction during production. Also, for stylistic consistency, the terms “survivors” and “non-survivors” could be written in lowercase throughout the manuscript.

Overall, this is a well-written, methodologically sound, and clinically valuable manuscript. The authors have clearly taken the reviewers’ feedback into account and refined the paper with care and precision. The study will be of clear interest to clinicians and researchers in cardiac surgery and perioperative medicine. I recommend acceptance after these very minor editorial adjustments.

 Response

Dear Reviewer,

Thank you for your valuable suggestions and comments, as well as your kind words about our manuscript! We truly appreciate the time and effort you have dedicated to evaluating our work.

Based on your suggestions, we made the required minor changes to the manuscript:

- The quotation marks around the institution’s name (“Prof. Dr. C.C. Iliescu”) were removed throughout the text and affiliations.

- We alphabetically ordered the notions of the Keywords section

- We completed the Conclusions with the suggested sentence regarding the role of VIS.

- We rearranged Figure 1 and the caption text in the manuscript.

- We checked the consistency of the terms “survivors” and “non-survivors” throughout the manuscript.

Round 3

Reviewer 1 Report

Comments and Suggestions for Authors

The adjustments made have been accepted